# Development of a Brief Coparenting Measure: The Coparenting Competence Scale

**DOI:** 10.3390/ijerph20136322

**Published:** 2023-07-07

**Authors:** Chris May, Codie Atherton, Kim Colyvas, Vincent Mancini, Linda E. Campbell

**Affiliations:** 1School of Health Sciences, Faculty of Health, University of Newcastle, Callaghan, NSW 2308, Australia; 2College of Engineering, Science and Environment, The University of Newcastle, Callaghan, NSW 2308, Australia; 3Curtin School of Population Health, Curtin University, Perth, WA 6102, Australia; 4School of Psychology, Faculty of Science, The University of Newcastle, Callaghan, NSW 2308, Australia

**Keywords:** coparenting, parenting, child behaviour, self-efficacy

## Abstract

Coparenting competence (CC) is a concept that describes the sense of collective efficacy that parents experience in raising children. An advantage of CC is that it bridges a gap between family systems thinking and efficacy theory, where extant research and theory have focused on the self-efficacy of one or both parents. This study aimed to develop a self-reported measure of CC. Methodology: Participants (*n* = 302), including cohabiting mothers (*n* = 240) and fathers (*n* = 62), completed an online survey (112 items) comprising demographic questions, the Coparenting Relationship Scale (CRS), the Parenting Sense of Competence Scale (PSOC), the Strengths and Difficulties Questionnaire (SDQ), and 36 items designed to explore perceptions of CC. Results: Factor analyses on 36-CC items identified 10 items that reliably formed a brief Coparenting Competence Scale (CCS; Alpha = 0.89). Analysis of convergent and divergent validity demonstrated that the CCS measures a unique construct that is linked to parenting self-efficacy, measured by PSOC (*r* = 0.47), and coparenting quality, assessed by the CRS (*r* = 0.63). There was a significant association between CCS and SDQ across age groups and an association stronger than that found for the CRS and SDQ in the current cohort. Conclusions and Implications: The study found support for the reliability and validity of the CCS. Coparenting competence, assessed by the CCS, was found to be distinct from factors previously used to represent coparenting quality in multivariate scales. The strength of associations between the CCS and SDQ suggests this new measure may have an important role in coparenting research.

## 1. Introduction

One of the most significant relationships in family systems is that which parents share in the raising of children [1]. This collaborative relationship is commonly referred to as the *coparenting relationship, parenting partnership, or the relationship that parents share in the business of raising children* [2]. Coparenting research typically explores the ways parents coordinate, communicate, support, and relate to each other on matters relevant to raising their children [1]. In addition to this dyadic relationship, coparenting theory and research also focus on children in a transactional triadic relationship wherein the personalities and behavioural characteristics of all members come into play [2]. Coparenting relationships are often explored in the context of biological parents but have been found to function independently of biological connections to the child, living arrangements, marital status, or sexuality [3,4,5,6,7]. The importance of addressing coparenting in family research and practice is increasingly recognised as evidence points to the influence that coparenting quality has on child and parent outcomes.

Coparenting research explores child and parent outcomes within the context of interactions between key members of the highly influential microsystem of the family environment, finding that the quality of coparenting is linked to both child and parent outcomes. Children are likely to exhibit fewer internalising and externalising behaviour problems, demonstrate enhanced emotion regulation, and perform better in relationships with their peers when they come from families where their parents demonstrate or report higher levels of quality in their coparenting relationship [8,9]. Coparenting quality is also linked to parent well-being, including parenting stress and marital satisfaction [10,11]. Indeed, studies exploring coparenting quality in mothers and fathers of typically and atypically developing children have reported that higher levels of coparenting quality predict lower levels of parenting stress (Delvecchio et al., 2015; May et al., 2015) [12,13]. Coparenting has also emerged as an important positive predictor of later marital behaviour, while the coparenting relationship is positively linked to marital quality and parenting practice [14]. 

Studies have also pointed to the malleability of coparenting relationships in response to targeted interventions. Brief interventions during the perinatal period have positively influenced coparenting quality and relationship functioning, with effects that can persist for at least two years post-intervention [15,16]. While research is mounting about the importance of coparenting quality for child and parent outcomes, studies are also exploring relations between coparenting quality and key parenting behaviours, such as mindfulness [17]. 

A supportive coparenting relationship has also been associated with self-reported parenting self-efficacy (PSE) in parents of typically and atypically developing children [13,18,19]. Parenting self-efficacy describes a parent’s estimation of personal competency in their parental role; alternatively, it is described as the parent’s perception of their ability to positively influence their child’s development and behaviour [20]. Parenting self-efficacy, a situation specific derivative of general self-efficacy, has a strong theoretical base, with accumulated evidence, pointing to the importance of PSE as a predictor and, for some elements, a determinant of factors such as parenting stress, parenting practices, strength of attachment relationships, child and adolescent behaviour, and coparenting quality [21,22,23,24]. While self-efficacy theory and research are often focused on individual characteristics, the concept can also be extended to explain the motivation and behaviour of groups where collective perceptions of efficacy predict both individual and group performance [25]. Although studies have explored aspects of group efficacy in the family system, such as marital and filial efficacy, research has not yet explored the perceptions of collective efficacy that are likely to occur in coparenting relationships. 

The sense of collective parenting efficacy that parents experience in their parenting relationship was first described by May et al as coparenting competence [26]. The concept of coparenting competence was developed during the qualitative component of May et al.’s study exploring the importance of coparenting quality in families raising a child on the autism spectrum. The concept of coparenting competence has the potential to bridge a gap between the collective perspective in family systems thinking and efficacy in parenting, which has, to this point, focused on the individual parenting self-efficacy of one or both parents. However, there are currently no available measures to assess perceptions of coparenting competence and the relationships that this factor could have with child and family outcomes. Instead, current measures assess an array of latent variables that are theoretically linked to coparenting and collectively provide an index of coparenting quality. For example, the most commonly used measures are the Coparenting Relationships Scale (CRS) [5] and the Parenting Alliance Measure (PAM [27]. The CRS has 7 subscales (Coparenting Agreement, Coparenting Closeness, Exposure to Conflict, Coparenting Support, Coparenting Undermining, Endorser Partner Parenting and Division of Labor) while the PAM explores 2 factors (Teamwork and Closeness). Although some items in these scales appear to focus on collective efficacy, the scales generally focus on the broader perspective of coparenting quality without exploring parental perceptions of the influence that their parenting relationship may have on the social and emotional development of their children. Consequently, a key component of the coparenting experience (i.e., collective parenting efficacy) remains underrepresented in existing assessments of coparenting [28]. 

### Aims and Objectives

The current study aimed to develop a measure of coparenting competence. The measure is differentiated from multifactorial measures of coparenting quality by focusing entirely on perceptions of collective coparenting efficacy. The development of such a measure will complement current measures exploring parenting relationships and support research exploring interventions designed to enhance coparenting competence in the hope of improving child and parent outcomes. 

The objectives were to create a reliable and valid measure of coparenting competence to evaluate the convergent and divergent validity of this measure against a commonly used multifactorial assessment of coparenting quality and a commonly used measure of parenting self-efficacy. The study will also test the ability of this new measure to predict parental perceptions of their children’s behavioural strengths and difficulties. 

## 2. Method

### 2.1. Study Design

An ‘open’ anonymous survey design was employed using the open-source survey tool LimeSurvey web application, supported by the University of Newcastle (UoN). The data were stored securely on a UoN server. A vulnerability check, completed by the UoN IT Assurance Team, concluded that the web application had adequate protection against unauthorised access. The survey was available online from February to August 2107. All procedures were approved by the University of Newcastle’s Human Research Ethics Committee.

### 2.2. Study Population

Our target group was parents aged ≥18 years who lived with their youngest child and partner for more than half the preceding year, whose youngest child was aged under 17, and who were sufficiently competent in written English to understand the study information, provide informed consent, and complete questionnaires.

### 2.3. Recruitment

A convenience sampling method was utilised with a broad recruitment strategy aimed at increasing the representativeness of the participants. Participants were recruited through online social media websites (e.g., Facebook and Twitter), programs offered to parents of preschool children at the Family Action Centre located at the UoN, word of mouth, snowballing (*n* = 200, 66.2% of total sample), and Prolific (*n* = 102, 33.8% of the total sample). The sample included 240 mothers and 62 fathers. Recruitment materials included brief blurbs, pamphlets, and posters outlining the purpose of the study and links to learn more about the study. Potential participants were directed to a study-specific website containing a URL link to the survey. A minimal incentive, an interactive parenting partnership workbook developed to support and enhance parenting partnership quality, was offered to all participants. Prolific offers demographic screening of participants and provides participants with a modest reimbursement for their time. Participants from Prolific provided a Prolific-specific identifier upon completing the survey; subsequently, the researchers advised Prolific that the participant had completed the survey, and they organised the transfer of reimbursement. No other identifying information was provided by Prolific, and no study-specific data were shared with Prolific. Prolific processes have been found to produce higher-quality data and more diverse participants than other popular recruitment services [29]. 

### 2.4. Procedure

All procedures were approved by the UoN Human Research Ethics Committee. When participants accessed the study website, they were presented with information about the study, including the names and contact information of the investigators, the purpose of the study, the approximate length of the survey, associated benefits and risks, and data storage and protection. Participants were asked to confirm that they had read the information and to indicate if they wished to proceed with the survey or not. Upon implied consent, the participants were provided access to the survey. The survey, whose usability and functionality had been fully tested, consisted of a brief demographic section to reduce participant burden (see Table 1 for complete demographic data), followed by four counterbalanced questionnaires described in the measures section below. The survey was distributed over 11 pages and took approximately 30 min to complete. Participants could change their answers by using a back button. They could also save responses and return to the survey later if they were unable to complete the full survey in a single session. Responses were mandatory with a visual alert indicating required items to be completed prior to the submission of the questionnaire. Participants could discontinue the survey at any time. All electronic responses were downloaded automatically into Excel or SPSS spreadsheets. 

### 2.5. Measures

#### 2.5.1. Coparenting Competence Scale (CCS)

Three experts, two male and one female, with academic expertise in psychology and coparenting, developed 53 potential items for the CCS. Experts were instructed to develop items, based on the conceptualization of coparenting competence. Items were therefore expected to explore a parent’s confidence in how well their parenting relationship adapted to their child’s evolving needs, how well their coparenting relationship functioned, and their expectations of the influence that their parenting relationship would have on their child’s developmental outcomes [26]. The lead author reviewed all items to ensure that expert suggestions were consistent with the stated aims by comparing each item to the previously described characteristics of coparenting competence. Items were then collated, sorted, and culled (when judged to be functionally identical or inconsistent with the characteristics of coparenting competence), leaving a corpus of 36 items. These items were then arranged on a survey wherein each could be assessed on a five-point Likert scale ranging from 0 (*not true at all*) to 4 (*very true*), with some designed for reverse scoring. The remaining items, presented in the previously described format, were then reviewed individually by a community reference group of parents (*n* = 7) for clarity and understanding. Parents in this group therefore reported on whether they found each item to be clearly stated and whether or not they believed that they understood the intended purpose of the item. Parents seldom reported concerns about the similarity among the items. Other concerns that were raised by parents were never consistent between two members on any occasion. All 36 items were therefore retained for the next stage of the study. 

#### 2.5.2. Coparenting Relationship Scale (CRS: Feinberg et al., 2012) [5] 

The CRS is a self-reported questionnaire consisting of 35 items loaded onto seven subscales (coparenting agreement, coparenting closeness, exposure to conflict, coparenting support, coparenting undermining, endorse partner parenting, and division of labour). All items are scored on seven-point Likert scales, with most ranging from 0 (*not true of us*) to 6 (*very true of us*), while the exposure to conflict subscale items range from 0 (*never*) to 6 (*very often*). Feinberg et al.’s longitudinal analysis of US data has demonstrated excellent internal consistency, according to Kline’s matrix, across gender and data collection time points [16,30]. Chronbach’s alpha (0.92) for total CRS has been demonstrated in Australian mothers and fathers, with similar results in Latham et al.’s (2018) English study using the short form survey [31].

#### 2.5.3. Parenting Sense of Competence Scale (PSOC; Johnston & Mash, 1989) [32]

The PSOC is a 16-item self-report measure of perceived parental competence, including two subscales: Satisfaction (*n* = 9 items), which includes items such as “Being a parent is manageable, and my problems are easily solved”; and Efficacy (*n* = 7 items) (e.g., “Sometimes I feel like I am not getting anything done”) with appropriate reversals. All items are scored on a six-point Likert scale ranging from 1 (*strongly agree*) to 6 (*strongly disagree*). Total scores on the PSOC are indicative of the respondent’s parenting confidence. The reliability of the PSOC in an Australian Study has shown alpha coefficients of 0.77 (mothers) and 0.80 (fathers) on the Satisfaction subscale, along with 0.78 (mothers) and 0.82 (fathers) on the efficacy subscale, with similar reliability demonstrated in English mothers and fathers [33,34].

#### 2.5.4. Strengths and Difficulties Questionnaire (SDQ; Goodman, 1997) [35]

Previous research has demonstrated significant relationships between reported coparenting quality using the CRS and both maternal and paternal perceptions of child behaviour [16]. The SDQ is a series of parent-rated, age-adjusted behavioural screening questionnaires for children aged 2 to 17 years. The SDQ was used in the present study to demonstrate the predictive validity of the CCS as compared to the CRS in relation to parental perceptions of child behaviour. Parents were asked to complete the SDQ as per the age of their youngest child. Parents (*n* = 8) who did not have a child 2 years of age or older were not offered the SDQ. Each survey consists of 25 items that form five subscales assessing perceived conduct problems, pro-social behaviours, emotional symptoms, peer relationship problems, and hyperactivity/inattention. Items are scored on a three-point Likert scale ranging from 0 (*not true*) to 2 (*certainly true*). The SDQ provides total scores for total difficulties and sub-scores for externalising and internalising behaviours. Higher scores indicate higher levels of difficulties, with the exception of the pro-social behaviour scale, where a higher score indicates strengths of the child. The SQD has been widely used, including in validation studies in Australian and UK samples [36,37]. 

### 2.6. Analyses

Analysis of key demographic data minimised the chance of data being included from single users filling in questionnaires multiple times. The two sources of the data (public recruitment and Prolific) were compared across a range of demographic variables and scale measures to determine if the samples were systematically different using *t*-tests, Mann-Whitney *U* non-parametric tests, and chi-squared tests as appropriate. The 36 items of the Coparenting Competence Scale (CCS) were assessed first with exploratory factor analysis (EFA) using Kaiser’s eigenvalue greater-than-one rule with maximum likelihood extraction and direct oblimin rotation [38]. 

An iterative process with one-at-a-time removal of poorer items (low loadings or high cross loadings) was used to reduce the scale to a measure including ten items. A total score was created by summing all ten items—negative items were reverse scored prior to analysis. Model acceptability for the measure was evaluated with confirmatory factor analysis (CFA), with goodness of fit assessed on an array of indicators, including Goodness of Fit, Tucker Lewis, and RMSEA. In addition, CFA was used to evaluate the psychometric properties of the CCS on the basis of the a priori assignment of items. Analyses were conducted with IBM SPSS and AMOS software (Version 22.0: SPSS, Chicago, IL, USA).

Cronbach’s alpha and composite reliability coefficients were calculated for all scales employed in the study. Convergent validity was assessed with correlation coefficients to determine the strength of theoretically derived associations between constructs of coparenting competence, coparenting quality, parenting self-efficacy, and perceptions of children’s externalising and internalising behaviours. The divergent validity between total scores was assessed by correction for attenuation, a process developed to ensure that the correlation between variables is estimated in a manner that accounts for measurement error [39]. The divergence between subscale scores was assessed with both attenuation and average variance extraction (AVE) [40]. Discriminant validity is supported by AVE when the product of the average variance extracted is greater than the square of the correlation between the subscales or factors.

## 3. Results

A total of 410 participants attempted the survey, with 108 excluded due to incomplete data. The remaining participants (*n* = 302) included 240 mothers (79.5%) and 62 fathers (20.5%), with an average parent age of 33.91 years (SD = 10.12). The average age of the youngest child in the family was 4.61 years (SD = 4.12). See Table 1 for further demographic data. 

Scale and demographic variables were examined for differences between those recruited from public sources and the Prolific service. None of the total scores for the scales examined in this study were significantly different between the two groups. The mean age of parent/caregiver and partner was significantly older (both *p* < 0.001) by about 5 years in the public group compared to the Prolific group. Similarly, the mean ages of the oldest and youngest children were higher (both *p* < 0.001) in the public sample by about 2.5 years compared to the Prolific sample. However, the mean number of children living with parents was not different. The proportion of females was higher (*p* < 0.001) in the public sample compared to Prolific (95% vs. 72%). Other demographic factors relating to living arrangements and the relationship with the partner were not significantly different. 

Exploratory Factor Analysis (maximum likelihood extraction and direct oblimin rotation) of responses to the initial 36 items developed for the scale revealed the presence of six components with eigenvalues exceeding 1, explaining 57.7% of the variance. An inspection of the scree plot revealed a clear break after the second component. An iterative reduction process, in which the item with the lowest maximum loading on any component was removed at each iteration, was then used to reduce the scale to a brief measure. The final two-component solution included 10 items explaining 70.5% of the variance, with Factor 1 contributing 55% and Factor 2 contributing 12.5%. The Kaiser-Meyer-Olkin Measure of Sampling Adequacy was 0.91 with satisfactory residual correlations from the reproduced correlation matrix (Range = −0.04 to 0.08, mean absolute deviation 0.02). The two factors were strongly correlated (*r* = 0.53, *p* < 0.01). Interpretation of these two factors identified positively worded items (*n* = 6) loading strongly on Factor 1 and negatively worded items (*n* = 4) loading strongly on Factor 2 (See Table 2). These factors are not sufficiently different, from a theoretical perspective, to warrant further definition, but they do serve a purpose in the exploration of a survey model that employs negative and positive items and in the analysis of average variance extracted to explore the similarity/difference between factors on the CCS from other surveys.

### 3.1. Factor Analysis

Confirmatory Factor Analysis (CFA; IBM SPSS AMOS, V24) using standardised estimates was performed to assess fit for a 2-factor solution including negative and positively scored items (Figure 1). Outcomes across all indices of fit indicated that the 2-factor model achieved acceptable fit in the current sample (see Table 3) [41,42]. Noting that factors are oblique correlations and therefore loadings can be larger than 1 [43]. 

### 3.2. Internal Consistency

The CCS demonstrated strong internal consistency (Cronbach’s Alpha = 0.89). Table 4 details Cronbach’s Alpha for the array of measures used in the present study by total and subscale scores. The CCS also demonstrated excellent internal consistency across identified genders with Cronbach’s alpha ranging from 0.84 (fathers) to 0.90 (mothers). 

### 3.3. Construct Validity

The CCS maintained a strong positive association with the total CRS (*r* = 0.63). The CCS also maintained a moderate to strong positive association with the brief CRS and all CRS subscales (*r* = 0.47 to 0.72) (see Table 5). The CCS had a moderate positive association with the PSOC total score (*r* = 0.47). These results demonstrate convergent reliability between the CCS and measures previously reported to reliably assess latent variables of coparenting quality and parenting self-efficacy. 

Correlations between the CCS and SDQ for age group measures were generally stronger than those found for the total CRS (*r* = −0.13 for 2–4 years, *r* = −0.13 for 5–10 years, *r* = −0.16 for 11–17 years); however, none were significant at *p* = 0.05. There was a significant difference (*p* = 0.02) when comparing relations between aggregated SDQ (2–17 years, *n* = 286) and CCS (*r* = 0.32) or CRS (*r* = 0.14). 

Correlations between the PSOC and SDQ (−0.332, −0.398, and −0.481) were all significant (*p* = 0.01). Correlations between total scores on the CCS, PSOC, and CRS were marginally stronger for women than men, while associations with the SDQ were skewed the other way (see Table 6). 

Correction of attenuation was performed on subscales that demonstrated reasonable reliability (alpha > 0.7) in the current sample. An outcome of <0.85 for attenuation is supportive of divergent validity (see Table 7).

Average variance extracted (AVE) was performed to determine discriminant validity between aggregated negatively scored and aggregated positively scored factors on the CCS and subscales of the PSOC and CRS. Discriminant validity is supported by AVE when the outcome is greater than the square of the correlation between aggregated items or factors [40]. Analysis supported discriminant validity between the aggregated CCS items and each of the subscales using both AVE and Spearman’s attenuation analysis (See Table 8) [39].

Attenuation analysis also supported discriminant validity between the CCS total score and total scores on all other measures (see Table 9) [39].

## 4. Discussions

The aim of this research was to develop and validate a quantitative measure of coparenting competence, a theoretically supported construct derived from qualitative inquiry [26]. The current study makes an important contribution to coparenting research with the development of the 10-item CCS and initial psychometric data revealing favourable indices of reliability and validity when compared to other theoretically related constructs. 

The present paper describes the development and reduction of items to form a short survey of coparenting competence. The study then went on to assess and demonstrate both the convergent and divergent validity (Campbell & Fiske, 1959) [42] of the resulting 10-item CCS with well-established measures of coparenting quality and parenting self-efficacy. These processes, along with the reliability of the CCS in the current sample, have demonstrated that the CCS assesses a unique variable that bridges the gap between coparenting quality and a collective sense of parenting efficacy. 

The construction of the CCS is founded on a distinct theoretical discourse that differentiates this measure from other multivariate measures of coparenting quality. This measure therefore provides a potentially important alternative to multivariate measures as key theorists exercise ongoing disagreement about an array of potential domains that constitute effective coparenting relationships [44]. The items on the CCS focus on parents’ beliefs regarding how well they work together in the business of raising children and their perceptions of the present and future influence that their collective parenting partnership has and will have on their child’s social and emotional development. The items on the CCS therefore assess perceptions of collective efficacy in relation to the functionality of the parenting partnership in both the way that they work together in parenting and the influence that this relationship will have on their developing child. The potential importance of coparenting competence has been illustrated in the link between efficacy and human agency, where perceptions of efficacy can be expected to predict how much effort is likely to be expended in sustaining and supporting the parenting partnership in the face of obstacles or aversive experiences [45]. 

Coparenting has been succinctly described as “how parents work together when they are raising a child” [46], and coparenting measures have primarily focused on factors that can be used to collectively represent how well this relationship is working. Unlike existing measures of coparenting quality or parenting alliance, which explore aggregated representations of the parenting partnership from factors theoretically linked to coparenting quality, the CCS assesses perceptions of collective efficacy in the specific context of parents’ ability to adapt to their child’s needs (e.g., “My partner and I are adapting well to meet our children’s changing needs”), work effectively together in the business of raising children (e.g., “I have a lot of confidence in our parenting partnership”), and collectively influence their child’s social and emotional development (e.g., “Our child/n is learning good relationship skills by seeing my partner and I work together in parenting”). This specific focus on collective parenting efficacy is not found in current multivariate coparenting measures. 

The theoretical underpinning of the CCS has been derived from qualitative research, founded on Bandura’s conception of collective efficacy [26,47]. The validity of this concept is now supported by empirical analysis demonstrating that coparenting competence can be differentiated from commonly used measures of coparenting quality and parenting self-efficacy. Although there is a strong correlation between the CCS and subscales on the CRS such as closeness (0.71, *p* < 0.001) and endorsement (0.72, *p* < 0.001), the analysis of discriminate validity consistently differentiates the CCS from CRS subscales. It is also notable, though not presented in the results, that similar correlations were also found between subscales on the CRS in the present study, such as undermining and agreement (*r* = 0.71, *p* < 0.001) and the subscales of closeness and support (*r* = 0.74, *p* < 0.001). It is important to note that the CCS is theoretically differentiated from the wider array of currently available coparenting measures that focus on specific circumstances (i.e., dissolved relationships, daily interactions, or feeding) or specific factors (i.e., triadic/dyadic parent interactions, interparental conflict, cooperation, and triangulation) [48].

It is also important to note that there are similarities between two items on the Parenting Alliance Measure (PAM) and items on the CCS. For example, one PAM item states that “My child’s other parent and I are a good parenting team”. Another item, “My child’s other parent makes my job of being a parent easier” also bears similarity to items on the CCS; for example, “Our child/ren would be easier to manage if I could do it on my own”. However, these two items do not explore perceptions of the influence that the relationship is likely to have on specific outcomes, and neither has been reported to assess collective parenting efficacy in this two-factor, 20-item scale (Konold & Abidin, 2001). The CCS uniquely and intentionally focuses on the construct of coparenting competence, which is the sense of collective efficacy that parents share in the business of raising children.

The development of this new measure may be particularly important due to the strength of associations between perceptions of coparenting competence and child behaviour in the present sample. It is well established that the quality of the parenting relationship is closely associated with child behaviour [49,50]. However, the current study demonstrated that the total CCS score was more strongly associated with parental perceptions of children’s strengths and difficulties (as measured by the SDQ) than total scores on the CRS. This suggests that perceived coparenting competence may interact with perceptions of child behaviour in different ways than factors assessed on the CRS. Hence, coparenting competence may have an important role in future studies exploring relationships between coparenting and perceived child behaviour. 

The CCS could also have an important role in assessing the efficacy of coparenting interventions. Interventions often focus on enhancing parental perceptions of competence in order to encourage parents to try harder and for longer to overcome the challenges that they face in their parenting work. The strength of association between the CCS and SDQ in the present cohort suggests that family researchers and clinicians should consider the potential importance of coparenting competence in intervention and evaluation.

## 5. Limitations

Limitations need to be acknowledged in the interpretation of these results. The self-selecting nature of the current sample may mean that responses are not representative of the entire population of parents, and caution needs to be exercised when generalising results. The socioeconomic position of parents and the number of parents who come from the same relationship is not known; with 26% of complete surveys coming from fathers, it is unlikely that the majority of surveys represent couple data. However, future studies will be needed to understand relationships between perceptions of coparenting competence within family systems. The study findings are also limited by the eligibility criteria (i.e., youngest child had to be under 17; the cohort were primarily biological cohabiting parents), which deliberately resulted in a reasonably homogenous cohort, and these results may not extend to other family types. Ideally the number of participants would have been more evenly distributed across genders; although the paternal data are encouraging, particular caution is needed when using the CCS with fathers.

The study also focused on coparenting competence related to the strengths and difficulties of only one child under 17, but the salience of CC dimensions may change as children grow older or as additional children are added to the analysis. As with the use of any self-report measure, the assessment of coparenting competence with the CCS is inherently subjective and may capture dimensions that arise using alternative methods. Findings from the present study will require additional validation in further studies. 

## 6. Conclusions

The concept of coparenting competence enables a link between efficacy theory and coparenting. The CCS provides a short measure of coparenting that is distinct from factors previously used to represent coparenting quality in multivariate measures. The psychometric analyses of the CCS yielded generally favourable results. The present study has added an empirical component to the development of the construct of coparenting competence, and the CCS adds to the field of coparenting research as a standalone measure focused on collective parenting efficacy. Future studies could use the CCS as a stand-alone measure of coparenting quality or as another strand in a multidimensional assessment.

## Figures and Tables

**Figure 1 ijerph-20-06322-f001:**
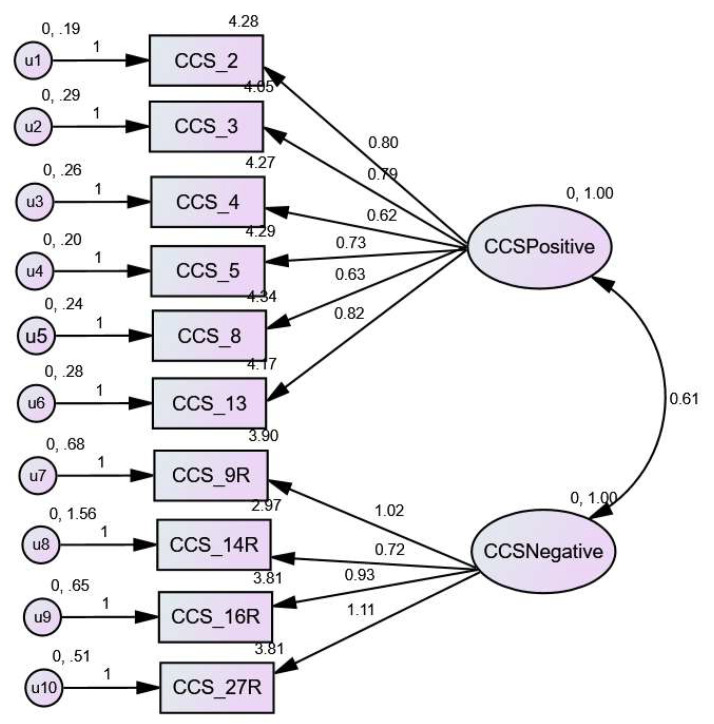
CCS 2 Factor Model. This figure illustrates the Coparenting Competence Scale (CCS) model with significant coefficients presented in standardised form. Note that AMOS does not generate 0 before decimal point in ranges on left of model.

**Table 1 ijerph-20-06322-t001:** Demographic Data (*n* = 302).

Child Age	Mean	SD	Range
Oldest child	7.99	4.98	0–38
Youngest child	4.60	4.12	0–17
All children	6.29	4.86	0–38
Number of children per family			*n* (%)
One			90 (29.8%)
Two			140 (46.4%)
Three			55 (18.2%)
Four			15 (5%)
Five			1 (0.3%)
Nine			1 (0.3%)
Relationship status			
Biological mother and father living together with youngest child.			278 (92.1%)
Blended family.			18 (6%)
One parent with biological child and partner.			3 (1%)
Adoptive or foster parents living together with child.			1 (0.3%)
Other			2 (0.7%)

**Table 2 ijerph-20-06322-t002:** Loadings (pattern matrix) and communalities for the Coparenting Competence Scale (CCS) with oblimin rotation—Exploratory Factor Analysis.

Item	Loadings—Pattern Matrix		
	Component 1	Component 2	Communalities
2 Our child/n is learning good relationship skills by seeing my partner and I work together in parenting.	**0.88**	0.00	0.77
5 Our parenting partnership is having a positive influence on our child/n’s behaviour.	**0.87**	−0.02	0.74
3 Our child/n is learning how to manage their behaviour and emotions by seeing how well my partner and I work together in parenting.	**0.84**	−0.01	0.69
4 My partner and I are adapting well together to meet our child/n’s changing needs.	**0.83**	−0.09	0.62
13 I have a lot of confidence in our parenting partnership.	**0.77**	0.12	0.70
8 My partner and I work well together as a parenting team.	**0.71**	0.13	0.62
16 Parenting works better when my partner does not interfere.	0.00	**0.77**	0.60
27 My partner makes parenting harder than it needs to be.	0.13	**0.75**	0.68
9 Our child/ren would be easier to manage if I could do it on my own.	0.08	**0.73**	0.61
14 I expected to work more closely with my partner in the parenting of our child/n.	−0.07	**0.56**	0.27

Note: Major loadings for each item in bold font.

**Table 3 ijerph-20-06322-t003:** CCS 2 factor model—CFA goodness of fit indices.

	Tucker Lewis Index	Normed Fitted Index (NFI)	Goodness of Fit Index (GFI)	Comparative Fit Index (CFI)	Root Mean Square (RMSEA)
Fit Index	0.95	0.95	0.934	0.97	0.08
Acceptability limit	>0.90 >0.95	0.90	0.90	0.93	0.08

**Table 4 ijerph-20-06322-t004:** Internal reliabilities of the 10-item Coparenting Competence scale (CCS), Coparenting Relationship Scale (CRS), Parenting Sense of Competence scale (PSOC), and the Strength and Difficulties questionnaire (SDQ) in the current sample.

Scale (Number of Items)	Cronbach’s Alpha
Coparenting Competence Scale (CCS; 10)	0.89
Coparenting Relationship Scale Total (CRS; 35)	0.77
CRS, brief version (14)	0.62
CRS, Coparenting agreement (4)	0.75
CRS, Coparenting closeness (5)	0.30
CRS, Exposure to conflict (5)	0.90
CRS, Coparenting support (6)	0.91
CRS, Coparenting undermining (6)	0.85
CRS, Endorse partner’s parenting (7)	0.61
CRS, Division of labour (2)	0.57
Parenting Sense of Competence Scale (PSOC; 16)	0.87
Strengths and Difficulties Questionnaire, ages 2–4 (SDQ; 25)	0.84
SDQ, ages 5–10 (25)	0.83
SDQ, ages 11–17 (25)	0.71

**Table 5 ijerph-20-06322-t005:** Correlations between Coparenting Competence Scale (CCS) total score and related constructs.

Scale (Number of Items)	Correlation with CCS
Coparenting Relationship Scale Total (CRS; 35)	0.630 **
CRS, Brief Version (14)	0.507 **
CRS, Coparenting agreement (4)	0.508 **
CRS, Coparenting closeness (5)	0.713 **
CRS, Exposure to conflict (5)	−0.587 **
CRS, Coparenting support (6)	0.683 **
CRS, Coparenting undermining (6)	−0.604 **
CRS, Endorse partner’s parenting (7)	0.719 **
CRS, Division of labour (2)	0.483 **
Parenting Sense of Competence Scale (PSOC; 16)	0.473 **
Strengths and Difficulties Questionnaire, ages 2–4 (SDQ; 25)	−0.319 **
SDQ, ages 5–10 (25)	−0.341 **
SDQ, ages 11–17 (25)	−0.294 *

Note: * Correlation significant at 0.05 ** Correlation significant at 0.01 (2-tailed).

**Table 6 ijerph-20-06322-t006:** Concurrent correlations between (CCS) and related constructs by gender.

Scale (Number of Items)	CCS	
	Female	Male
Coparenting Relationship Scale Total (CRS; 35)	0.643 **	0.537 **
Parenting Sense of Competence Scale (PSOC; 16)	0.481 **	0.458 **
Strengths and Difficulties Questionnaire, ages 2–4 (SDQ; 25)	−0.317 *	−0.319
SDQ, ages 5–10 (25)	−0.318 **	−0.435 *
SDQ, ages 11–17 (25)	−0.275 *	−0.548

Note: * Correlation significant at 0.05 ** Correlation significant at 0.01 (2-tailed).

**Table 7 ijerph-20-06322-t007:** Attenuation analysis between Coparenting Competence Scale (CCS) and Coparenting Relationship Scale (CRS).

Scale	Cronbach’s Alpha	Correlation with CCS	Attenuation Score
Total (CRS)	0.801	0.630	0.748
CRS, Brief Version	0.615	0.507	N/A
CRS, Coparenting Agreement	0.753	0.580	0.710
CRS, Coparenting Closeness	0.301	0.713	N/A
CRS, Exposure to Conflict	0.895	−0.587	−0.659
CRS, Coparenting Support	0.911	0.683	0.760
CRS, Coparenting Undermining	0.845	−0.604	−0.698
CRS, Endorse Partner Parenting	0.719	0.608	0.981 *
CRS, Division of Labour	0.567	0.567	N/A

Note: N/A if alpha < 0.7. * This subscale failed a test of convergent validity (Average Variance Extracted = 0.45) which supports discriminate validity.

**Table 8 ijerph-20-06322-t008:** Analysis of discriminant validity.

Scale/Subscale 1 (x)	Scale/Subscale 2 (y)	Average Variance Extracted Valid When AVEx*AVEy > *r*^2^		Attenuation = *r*^2^ /(αx*αy) Valid If <0.85		
		*r* ^2^	(AVEx*AVEy)	αx	αy	*r*^2/^(αx*αy)
CCS 4 (neg)	CRS, Support	0.25	0.56	0.80	0.90	0.35
CCS 6 (pos)	CRS, Support	0.26	0.64	0.94	0.80	0.35
CCS 4	CRS, Conflict	0.20	0.61	0.79	0.92	0.27
CCS 6	CRS, Conflict	0.28	0.71	0.94	0.93	0.33
CCS 4	CRS, Endorse	0.41	0.51	0.80	0.88	0.58
CCS 6	CRS, Endorse	0.38	0.59	0.94	9.87	0.46
CCS 4	CRS, Closeness	0.30	0.47	0.80	0.80	0.48
CCS 6	CRS, Closeness	0.47	0.58	0.94	0.80	0.62
CCS 4	CRS, Agreement	0.27	0.48	0.80	0.76	0.44
CCS 6	CRS, Agreement	0.24	0.58	0.94	0.76	0.59
CCS 4	CRS, Undermining	0.40	0.59	0.80	0.86	0.58
CCS 6	CRS, Undermining	0.18	0.61	0.94	0.86	0.23
CCS 4	CRS, Exposure	0.24	0.61	0.80	0.92	0.32
CCS 6	CRS, Exposure	0.28	0.71	0.94	0.92	0.33
CCS 4	PSOC, Satisfaction	0.18	0.42	0.80	0.81	0.28
CCS 6	PSOC, Satisfaction	0.16	0.53	0.94	0.81	0.21
CCS 4	PSOC, Efficacy	0.02	0.46	0.80	0.84	0.37
CCS 6	PSOC, Efficacy	0.16	0.57	0.94	0.84	0.20

Note: *r* = correlation between subscales and α = *composite reliability*. CCS = Coparenting Competence Scale, CRS = Coparenting Relationship Scale, PSCOC = Parenting Sense of Competence.

**Table 9 ijerph-20-06322-t009:** Attenuation analysis between the CCS and related scales.

Scale 1 (x)	Scale 2 (y)	Attenuation = *r*^2^ /(αx*αy) (Valid If <0.85)			
		*r*	αx	αy	*r*^2/^(αx*αy)
CCS	CRS	0.63	0.89	0.80	0.56
CCS	PSOC	0.47	0.89	0.87	0.28
CCS	SDQ (2–4)	−0.32	0.89	0.85	0.13
CCS	SDQ (5–10)	−0.34	0.89	0.83	0.16
CCS	SDQ (11–17)	−0.29	0.89	0.71	0.13

Note: *r* = correlation between subscales and α = composite reliability. CRS = Coparenting Relationship Scale, PSOC = Parenting Sense of Competence, SDQ = Strength and Difficulties Questionnaire.

## Data Availability

Data available on request from corresponding author due to privacy concerns.

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
