# Peer review of "Development of a Brief Coparenting Measure: The Coparenting Competence Scale"

_ijerph, 2023, doi:10.3390/ijerph20136322_

Round 1
Reviewer 1 Report
Intro
Well written informative intro, although the paragraph from line 41 to 53 has no single citation but refers to previous research on coparenting
Methodology
Measures
Line 61 states- “the led author reviewed all items to ensure that expert suggestion were consistent with the stated aims”- aims of? Also, what was done? How did the lead author ensure that?
Line 71- parents concerns on items need more clarity, not really well explained what was done and how?
Analysis and Results sections are strong and provide detailed description of what has been done
Discussion section needs to highlight the study findings but better contrasting and comparing it with previous research, e.g., line 373 I assume authors refer to their study, filling the gap and providing empirical evidence on differentiation of coparenting quality and self-efficacy.
Author Response
Thank you to the reviewers for the time and effort that they have invested in this paper. The comments have been extremely helpful and contributed to the quality of the manuscript.
Please note that all changes to the document, including additional changes inspired by the review process are highlighted
We have listed our responses (indented) under each of the reviewer comments.
Reviewer1
Intro
Well written informative intro, although the paragraph from line 41 to 53 has no single citation but refers to previous research on coparenting
Multiple citations have been added to this and other aspects of the introduction.
Methodology
Measures
Line 61 states- “the led author reviewed all items to ensure that expert suggestion were consistent with the stated aims”- aims of? (further information added) Also, what was done? How did the lead author ensure that?
Further information has been added to elaborate on what was done here. This was simply a check to ensure that items fitted with the theoretical conception of coparenting competence derived from the lead author’s qualitative research.
Line 71- parents concerns on items need more clarity, not really well explained what was done and how?
Further information added to explain that parents were reviewing items for readability and clarity (that they were easy to read and that the meaning seemed to be clear)
Analysis and Results sections are strong and provide detailed description of what has been done
Discussion section needs to highlight the study findings but better contrasting and comparing it with previous research, e.g., line 373 I assume authors refer to their study, filling the gap and providing empirical evidence on differentiation of coparenting quality and self-efficacy.
Thank you. This has been an important contribution to the paper. Further information has been added regarding differentiation of CCS from the CRS and other coparenting measures.
Reviewer 2 Report
Dear Authors,
The paper sounds interesting, and I trust it will add to the existing knowledge. Here are some remarks to consider in revising the manuscript:
Introduction
- Several parts in the introduction need citations. The authors state some views/conclusions without support. For example, p. 1, lines 30-35), all these need citations. Another example is the start of the second paragraph in the introduction (line 41) which has no citations. This goes on until the end of the introduction. The researchers should carefully revise these parts and add relevant resources that support these views.
- P. 2, line 39: It is helpful if the subscales of these instruments are briefly mentioned and explained to help the reader identify the gaps.
Methodology
- P. 3, line 14: insert the missing dates.
- P. 3, line 19: who set this criterion and why “for more than half the preceding year”?
- In the recruitment section, no information is available about sex, SES, and other important demographic variables.
- P. 4, for the CRS and PSOC scales, there should be more studies related to the reliability and validity of these scales. The reader wants to know if they have any psychometric properties in the context where the study was conducted.
- P. 5, end of first paragraph: The SDQ was validated in Australia, but what about the context of the study (UK)?
- P. 5, line 11: It’s good to mention the EFA relevant statistics.
- P. 5, line 16: Are there any model fit indices to report?
- Some of the results in the text don’t have significance levels (e.g., P. 6, line 59). The reader has to go back to the tables to look at the significance levels.
- P. 7: Table 2 should indicate that this is an EFA.
- I am not sure why the authors chose SDQ in finding the concurrent correlation between CCS and related constructs. Which type of validity is this? If this is concurrent (criterion-related) validity, then in what was are the subscales/constructs in these scales are similar and comparable.
Discussion
Overall acceptable but needs more support with literature in relation to the psychometric properties and the authors need to discuss how the results of the study corroborate with those of other researchers in different contexts.
Author Response
Dear Authors,
The paper sounds interesting, and I trust it will add to the existing knowledge. Here are some remarks to consider in revising the manuscript:
Thank you for the encouragement.
Introduction
- Several parts in the introduction need citations. The authors state some views/conclusions without support. For example, p. 1, lines 30-35), all these need citations. Another example is the start of the second paragraph in the introduction (line 41) which has no citations. This goes on until the end of the introduction. The researchers should carefully revise these parts and add relevant resources that support these views.
Additional, relevant citations as per this and other reviewer comments
- P. 2, line 39: It is helpful if the subscales of these instruments are briefly mentioned and explained to help the reader identify the gaps.
Subscales now included
Methodology
- P. 3, line 14: insert the missing dates.
Dates added
- P. 3, line 19: who set this criterion and why “for more than half the preceding year”?
This criteria was set arbitrarily be the investigators as a way of providing a context for the type of coparenting arrangements that the parents in this study were sharing. The aim being to explore the validity of this measure in cohabiting parents.
- In the recruitment section, no information is available about sex, SES, and other important demographic variables.
Minimal demographic data was collected to reduce participant burden and to ensure that only relevant data was obtained. The complete demographic data, other than gender is presented in table 1. Data on gender of participants is presented in the results section. Gender is now included in the recruitment section.
- P. 4, for the CRS and PSOC scales, there should be more studies related to the reliability and validity of these scales. The reader wants to know if they have any psychometric properties in the context where the study was conducted.
(Data on reliability of in CRS and PSOC in contemporary Australian and UK studies now included0
- P. 5, end of first paragraph: The SDQ was validated in Australia, but what about the context of the study (UK)?
- P. 5, line 11: It’s good to mention the EFA relevant statistics.
Information added re eigenvalue
- P. 5, line 16: Are there any model fit indices to report?
Examples of model fit indices now reported in method section.
- Some of the results in the text don’t have significance levels (e.g., P. 6, line 59). The reader has to go back to the tables to look at the significance levels.
P value added to P6 line 59. Document searched for other examples and rectified when found.
- P. 7: Table 2 should indicate that this is an EFA.
- This has been attended to
- I am not sure why the authors chose SDQ in finding the concurrent correlation between CCS and related constructs. Which type of validity is this? If this is concurrent (criterion-related) validity, then in what was are the subscales/constructs in these scales are similar and comparable.
The SDQ has been employed to demonstrate predictive validity. This is now clearly explained and referenced in the method section.
Discussion
Overall acceptable but needs more support with literature in relation to the psychometric properties and the authors need to discuss how the results of the study corroborate with those of other researchers in different contexts.
It is not possible to compare these results with others due to the pioneering nature of the research. However the authors have further developed the arguments in the discussion section to demonstrate differentiation between the CCS and the broader stable of coparenting measures and the importance of a measure that does not rely on the contested array of multivariate factors currently used to represent coparenting quality.
Reviewer 3 Report
Dear Authors,
I really enjoyed the research. I am going to make a series of comments for its improvement. First of all, please number the sections. Second, add hypotheses and explain how they are confirmed or refuted. Third, review the section on practical applications and complement the discussion with meta-analysis studies.
Author Response
I really enjoyed the research. I am going to make a series of comments for its improvement. First of all, please number the sections. Second, add hypotheses and explain how they are confirmed or refuted. Third, review the section on practical applications and complement the discussion with meta-analysis studies.
Thankyou for the encouragement.
Thank you also for the time and effort invested in this paper. The comments have been extremely helpful and contributed to the quality of the manuscript.
Please note that all changes to the document, including additional changes inspired by the review process are highlighted in the revised manuscript.
Hypotheses are not usually stated in scale development research papers. However, we believe that the manuscript has been improved by adding a heading Aims and Objectives to highlight these more clearly to the reader.